# Superconductivity in Hierarchical 3D Nanostructured Pb–In Alloys

**Artem F. Shevchun** [1,2,*] , **Galina K. Strukova** [1] , **Ivan M. Shmyt'ko** [1] , **Gennady V. Strukov** [1] , **Sergey A. Vitkalov** [3] , **Dmitry S. Yakovlev** [4,5,*] , **Ivan A. Nazhestkin** [4,5] and **Dmitry V. Shovkun** [1,2]

[1]  Institute of Solid State Physics, Russian Academy of Sciences, 142432 Chernogolovka, Russia
[2]  Joint Department of Condensed Matter Physics, HSE University, 101000 Moscow, Russia
[3]  Physics Department, The City College of New York, New York, NY 10031, USA
[4]  Russian Quantum Center, Skolkovo, 143025 Moscow, Russia
[5]  Center for Advanced Mesoscience and Nanotechnology, Moscow Institute of Physics and Technology, 141700 Dolgoprudny, Russia
*  Correspondence: shevchun@mail.ru (A.F.S.); dmitry.yakovlev@phystech.edu (D.S.Y.)

**Abstract:** The superconducting properties of hierarchical nanostructured samples of Pb–In alloys have been studied by the measurement of dynamic susceptibility $\chi(T)$ temperature dependence. Symmetric samples with different shapes and sizes were formed on a brass metallic net by cathode-metal electrodeposition with a programmed pulsing current. Two different kinds of $\chi(T)$ dependence were observed in synthesized structures. The first kind was a broad superconductive transition without energy dissipation with a very weak response to the external magnetic field. The second kind was, conversely, an abrupt transition signifying an energy dissipation with a significant field response. This behavior depends on the ratio between a superconducting domain size (defined by the London penetration depth $\lambda$) and a crystallite size. In these cases, one or several superconducting domains are present in a sample. This result paves the way to controlling a superconducting domain size in materials with the parameters of a pulsed current.

**Keywords:** superconductivity; Pb–In alloy hierarchical structures; nanocrystal; dynamic magnetic susceptibility; magnetic field penetration depth; size effect

## 1. Introduction

Zero-dimensional (0D systems) [1], one-dimensional (1D systems) [2], or two-dimensional (2D systems) [3] dimensions are low-dimensional systems. Quantum dots, quantum wells, and quantum nanowires are popular names for 0D, 1D, and 2D structures in the Society of Condensed Matter and Solid-State Physics, respectively. The usage of closed structures is also common in many other scientific domains, including thermoelectric materials [4], metamaterials [5], topological materials [6], energy storage applications [7], and catalysis [8]. Since confinement typically takes place at the nanoscale scale in these applications, low-dimensional systems are typically referred to as nanostructures.

As is well known, the parameters of both a crystal lattice and electrons change significantly when the system sizes reduce to a nanometer scale, where the size effects related to interruptions of the lattice periodicity by sample boundaries, as well as the discreteness of the electron quantum spectrum, play a significant role. The superconducting parameters of nanoscaled electron systems under zero-dimensional (0D) conditions [1], i.e., when all system sizes are considerably less than characteristic superconductivity lengths, substantially differ from those of bulk materials. For example, materials with dimensions smaller than a superconductor coherence length $\xi$ lead to phase fluctuations and topological defects [9]. An ultimate size effect in superconductors is a complete disappearance of superconductivity when electron-level separation near the Fermi level becomes comparable with the bulk superconducting gap [10]. According to the Anderson criterion, it happens when the particle sizes are a few nanometers [11,12].

The low-dimensional superconductors have a wide variety of applications. Superconducting nanowires may be used as microwave radiation detectors, as a base for quantum qubits [13] and single-photon detectors [14]. Nanoscale superconductors grown on semiconductor substrates are promising elements for superconducting electronics [15]. The study of various proposals, the low-dimensional SC in nanowires with strong spinorbit interactions, has stimulated great enthusiasm in exploring the physical properties of topological superconductors [16]. Superconducting aluminum nanoparticles, surrounded by an oxide layer, are promising material for quantum information processing circuits due to their large kinetic inductance [17]. Thus, the investigation of their superconducting properties at different sizes is a current task.

For low-dimensional superconductors, many investigations have been performed on thin films and nanowires. The quench of superconductivity below a critical film thickness [18] and superconducting transition temperature ($T_C$) oscillations with film thickness have been studied in Pb films [19], as well as in Al films [20]. A suppression of superconductivity was observed with a diameter of Al nanowires [13,21]. For 0D superconducting nanoparticles, superconductivity modulation with particle size was also observed. Measurements of the magnetic susceptibility of Pb nano-powders are presented in [22,23]. For particles with a size of less than 6 nm, the size effects become significant. For these nanoparticles the temperature of the superconducting transition is reduced, while the critical magnetic field increases abruptly. In particular, the temperature of the superconducting transition of 4.5 nm-particles reduces to 4.6 K, and the critical magnetic field is at least two orders of magnitude higher than that in a bulk Pb. No signs of superconductivity were observed for particles with a size of less than 2 nm. In [24], for Pb nanoparticles, the critical temperature decreased with particle size according to the exponential law. An interesting peculiarity was found for In nanoparticles with tetragonal lattice symmetry: the temperature $T_C$ increases by 5% with a decrease in particle size before the particle size reaches the Anderson limit [25,26]. For example, the novel class of two-dimensional hexagonal superconductors show a drop in $T_C$ of 2D monolayer $CaC_6$ as compared to its bulk equivalent, but an increase in $T_C$ when monolayer $LiC_6$ as compared to its bulk form [27]. Materials' critical temperature values are influenced by non-adiabatic effects due to similar electron and phonon energy scales at low dimensions. In particular, the other materials might react similarly to $LiC_6$ and subsequently have their $T_C$ reduced as a result [28]. As a result, the articles mentioned above help us comprehend the underlying physics of low-dimensional superconductivity and present new avenues for further study.

This brief review indicates that low-dimensional superconductors such as thin films, nanowires, and nanopowders serve as convenient objects for studying the effect of size on superconductivity. Recently, a novel class of materials has been developed containing hierarchical 3D nanostructures [29] with a complex symmetry nano-architecture [30–32]. What are the superconducting properties of these complex 3D nanostructured materials? This important question is still open. In this article, we investigate the relation between the structure and superconducting properties of the novel 3D hierarchical nanoscaled materials using dynamic magnetic susceptibility as the method to estimate the effective size of superconducting domains screening the magnetic field. We present a study of nanostructured Pb–In systems, which have a complex hierarchical order from hundreds of micrometers down to a few nanometers. The geometry and structure of the 3D hierarchical objects are studied by a scanning electron microscope (SEM) and X-ray phase analysis. The studied samples contain objects of different sizes: nanocrystals having the smallest size of 10–100 nm, crystallites—100 nm–1 μm, coupled with one another in different ways, forming complex structures which measure up to 100 μm. We used contactless comparative measurements of a temperature dependence of the dynamic magnetic susceptibility at a frequency of 100 kHz as a sensitive instrument characterizing the response of superconducting samples to the external magnetic field. Recently, Riminucci and Schwarzacher [33] have demonstrated the possibility of estimating the sizes of the superconducting domains in polycrystalline arrays of Pb rods of 200 nm in diameter using magnetization data of the samples. Below, we made estimates of the characteristic size of unconnected supercon-

ducting regions based on the measured temperature dependence of the dynamic magnetic susceptibility.

## 2. Methods

Samples are fabricated using cathode metal electrodeposition on templates in an aqueous electrolyte according to the effective method for the fabrication of hierarchical nanostructured 3D systems [29,34]. All experiments are conducted in a tight-lid growth cell equipped with a thermometer and vent tube, as shown in Figure 1a. A brass metallic net with a size of $10 \times 30$ mm$^2$ coated with a polymeric porous membrane served as a cathode. The mesh size was $150 \times 150$ µm with a wire diameter of 100 µm. A plate, fixed at a distance of 1 cm from the polymeric membrane, served as an anode. For all samples, a lead cathode was used, except for the Pb–In–Pd 93/4/3 sample, when the palladium cathode was used. The cathode and anode were immersed in the electrolyte, and a programmed pulsing current passed through the solution. Depending on a growth mode, structures of different compositions and forms were grown on the membrane Figure 1b. Samples of the composition of Pb–In 65/35 (with indium content of 35), Pb–In 85/15, and Pb–In–Pd 93/4/3 were obtained from the electrolyte heated to 50–60 °C and containing (g/L): $PbCl_2$-15; $InCl_3$-50; $NH_4Cl$-100. The block scheme of the experimental setup is presented in Figure 1c. The electrodeposition time was 650, 270, and 225 s at the amplitude of a rectangular current pulse of 300, 167, and 133 mA, frequency of 38, 167, and 167 Hz, and pulse ratio 50, respectively. We use the pulse current generator model 508 with the ability to generate precise current pulses. To control the current pulses, we used an Agilent DSOX3012A digital oscilloscope (200 MHz).

SEM studies of the samples were performed on SUPRA 50VP and JEM-2100 scanning electron microscopes. For this to be done, the structures grown on the cathode were washed with acetone followed by alcohol in an ultrasonic bath. A drop of the alcohol suspension was applied to a carbon-coated copper netting. After evaporation of the alcohol, the structures were placed on an electron microscope column. The X-ray phase analysis of the samples was performed on a Siemens D-500 diffractometer with a graphite post-monochromator using Cu K$\alpha$ radiation. The most successful method of preparing samples was to pipe the sample directly onto a small circular coverlip, fix the coverlip vertically in a sample holder, and collect a sequence of oriented scans.

Measurements of the dynamic magnetic susceptibility $\chi$ of the samples were performed using the method described in [35]. In brief, the dynamic magnetic moment M of the sample was measured in the external alternating magnetic field $H(t) = h\cos\omega t$. It yields the dynamic magnetic susceptibility $\chi$ via the relation $M = \chi V h$, where $V$ is the superconducting volume of studied samples. At the transition temperature $T_C$, the diamagnetic transition of a sample is observed on a $\chi(T)$ curve, and $T_C$ defined using this method shows good agreement with $T_C$ defined with an electrical resistivity $\rho(T)$ measurement [36]. In our experiment, a part of the grown structure with a mass m ~1 mg was gathered into a small Teflon container. The sample was placed between a pair of coaxial coils with a diameter of 6 mm. One coil served for excitation of an alternating magnetic field, and the other one was used for the measurement. An identical pair of coils located nearby is necessary for signal compensation in the absence of a sample. An imbalance signal was measured using a standard synchronous detection circuit. This signal is proportional to the magnetic moment $M = \chi V h$, where $h \approx 0.1$ Oe is the amplitude of the magnetic field produced by a coil, V is the volume of the sample. Measurements were performed at a magnetic field frequency of 100 kHz in a temperature range between 4 K and 300 K.

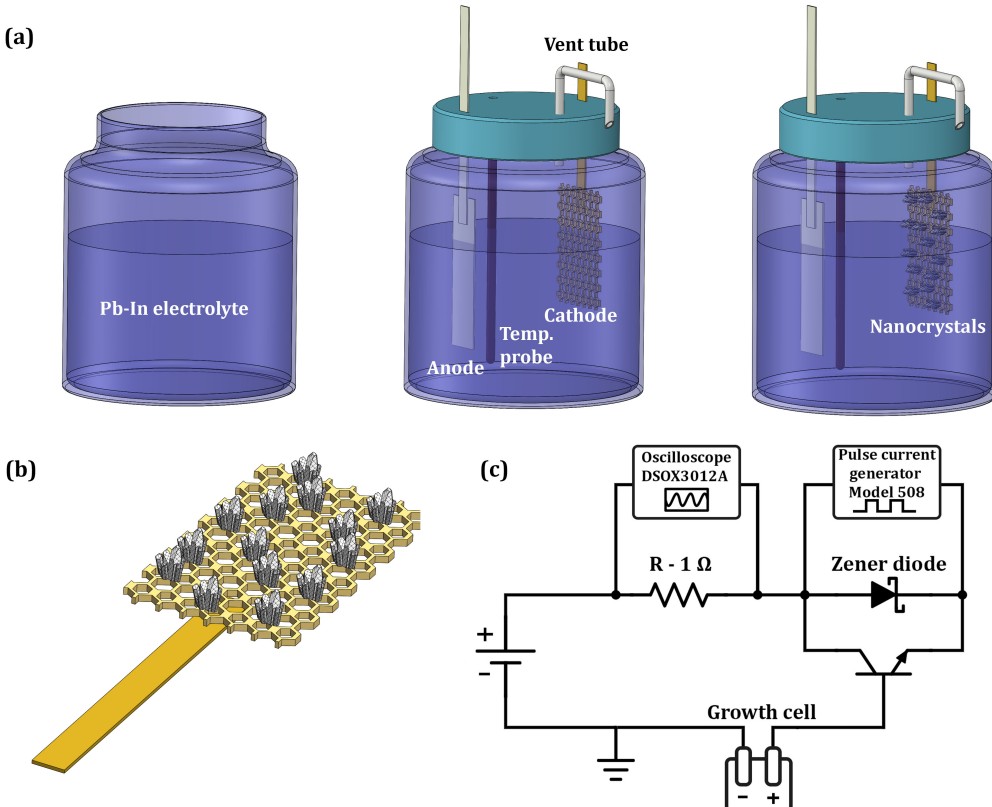

**Figure 1. Hierarchical 3D nanostructured Pb−In sample growth.** (**a**) Schematic illustration for the growth procedure for 3D nanostructured Pb−In samples and electrodeposition cell; (**b**) Schematic view of the cathode membrane covered by Pd−In nanocrystals; (**c**) The block-scheme of the experimental setup.

## 3. Results and Discussion

### 3.1. Sample Structures

To understand the surface morphology and detailed features of the nanocrystals, SEM was performed as shown in Figure 2. The Pb–In 65/35 samples (Figure 2a) have volumetric mesoporous structures obtained via a self-organization chain comprised of nanocrystals links. In the chain array shown in Figure 2a, there are straight and curved chains with a cross-sectional size of a few microns and a length of up to 100 μm. The chains consist of micrometer-sized crystallites. The chains can bend and do not rupture under treatment with alcohol suspension in the ultrasonic bath. In the bottom picture in Figure 2, one can see that the crystallites contain nanocrystals with a size of 50–100 nm.

These samples show an unusual X-ray spectrum in Bragg–Brentano geometry (Figure 3). One can see that each reflection splits into two components indicating a presence of two phases in the sample. According to the structural database PDF-2, these phases have a space symmetry group Fm3m (225). The X-ray measurements demonstrate the same angular dependence of the reflection indexes corresponding to different phases. This proves that the crystal lattices of the phases are isomorphic. The presence of two isomorphic phases contradicts to the phase diagram according to which only one phase should exist for indium and lead compositions. The contradiction is relaxed if one assumes that the particles formed during the initial synthesis stage have a 'core-shell' structure, i.e., they consist of a shell phase and a core. Previously, we have found such structures in a number of oxide and fluoride compounds of rare earth metals [37,38]. To determine the sizes of the shell phase and the core, we used the Selyakov method [39,40], substituting the value of half-width $\beta$ of different reflections into the formula: $\beta = \frac{\lambda}{D \cos \Theta}$. The average size of the nanoparticle core ($D_{inner}$) and the thickness of the shell phase ($D_{surf}$) calculated using the Selyakov formula for different reflections are $60 \pm 10$ nm and $25 \pm 5$ nm, respectively. Note

that these values obtained from the X-ray data agree with the sizes of particles observed in SEM images (Figure 2a).

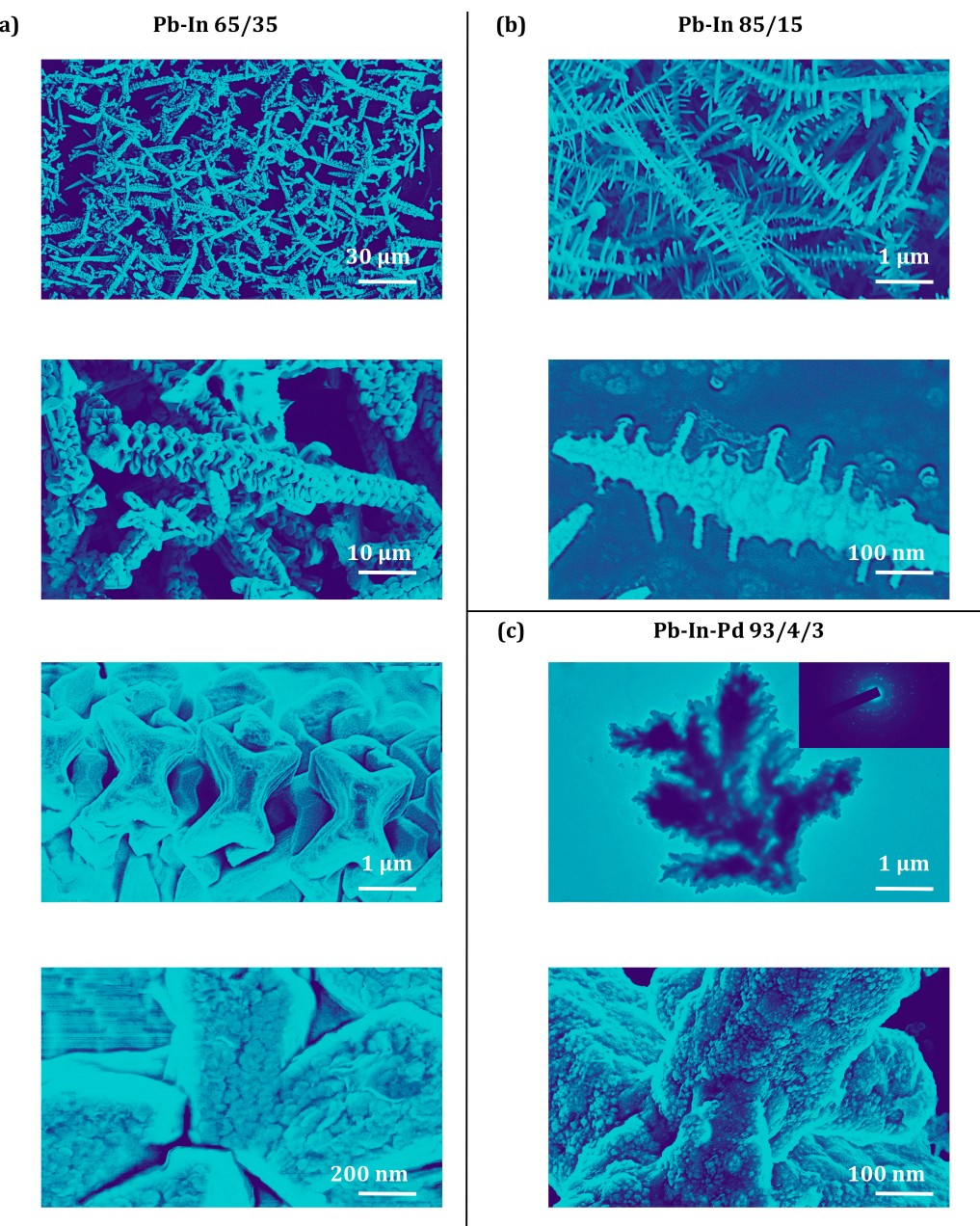

**Figure 2. Summary of the growth samples with different concentration of Pb and In.** (**a**) SEM images of the samples of Pb–In 65/35 alloy include: single chain, bulk chain links—crystallites and separate link, i.e., crystallite, consisting of nanocrystals; (**b**) SEM images of the nanorods of Pb–In 85/15 alloy include a general view and single-nanorod view; (**c**) SEM images of Pb–In–Pd 93/4/3 alloy include structure fragments, electron diffraction pattern (insert), and crystallite-containing nanocrystals.

Changing the regimes of the pulsing current, we obtained the hierarchical Pb–In 85/15 mesostructures, which have the form of barbed wire and consist of rods with extensions (Figure 2b). Crystallites in these mesostructures have characteristic sizes of 100–300 nm. The crystallites contain 10–30 nm-sized nanocrystals. The reduction in nanocrystal sizes in these samples is related to a significant increase in the frequency of current pulses during electrodeposition.

The grain refinement in these samples can also be achieved using a palladium anode. This is related to the palladium dissolution during electrolysis leading to palladium precipitates. Figure 2c presents SEM images of the Pb–In–Pd 93/4/3 sample. The branch-like, hierarchical meso-structure contains crystallites of 200–300 nm in size. The electron diffraction pattern shows that the sample consists of nanocrystals. Figure 2c shows that the nanocrystal size is about 10 nm.

The Pb–In material 3D nanostructures described in this section are new systems exhibiting complex symmetry architectures, including a volumetric net with angles and pores, composition, and structural inhomogeneity, which includes a complex symmetry structure. According to the contemporary understanding [41], these nanostructures with hierarchical nanoarchitectures can exhibit new properties and effects, providing further development opportunities based on quantum properties of matter.

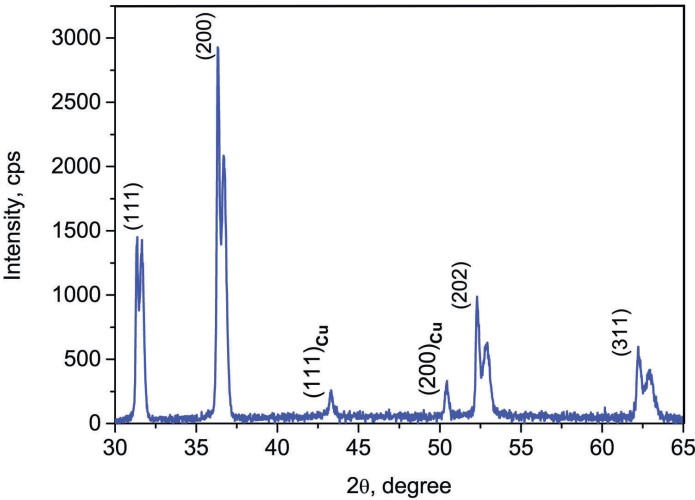

**Figure 3.** A representative XRD pattern recorded for the micro-chains of Pb–In 65/35 alloy. The peaks corresponding to diffraction on different atomic planes are labeled with (hkl) indices.

*3.2. Dynamic Magnetic Susceptibility of Pb–In Nanocrystalline Samples*

3.2.1. Theory

In the general case, the dynamic susceptibility $\chi = \chi' + i\chi''$ is a complex value that depends not only on the material of a sample, but also on its size and shape. The imaginary part $\chi''$ is proportional to the dissipation of the energy of the electromagnetic field in a sample, while the real part $\chi'$ is determined by screening the magnetic field by the sample. In the case of a spherical sample, the susceptibility depends on parameter $D/\lambda$, which is the ratio of the sample diameter $D$ to the depth of magnetic field penetration $\lambda$. The temperature dependence of susceptibility $\chi(T)$ is determined by the temperature dependence $\lambda(T)$. For large samples (the case $D \gg \lambda$), susceptibility is a step function abruptly changing at the superconducting transition temperature $T_C$. For small samples (the case $D < \lambda$), the susceptibility is proportional to $\left(\frac{D}{\lambda(T)}\right)^2$ and smoothly depends on the temperature even if the sample becomes fully superconducting below $T_C$. An exact formula for the spherical superconducting sample is obtained in [42]:

$$\chi'(t, D) \sim 1 - \frac{6\lambda(T)}{D}\coth\frac{D}{2\lambda(T)} + 12\frac{\lambda(T)^2}{D^2}. \tag{1}$$

In the intermediate case (the case $D \approx \lambda$), the dependence is more complex and depends on the shape of the sample. For inhomogeneous samples, a temperature dependence of the superconducting volume of a sample, $V(T)$, is an additional important factor

for the case where different parts of the sample undergo a superconducting transition at different temperatures. This property is particularly important for HTSC superconductors with a short coherence length. For ordinary superconductors, such as Pb–In alloy, with a long coherence length, all parts of a sample undergo the superconducting transition practically simultaneously, and we can neglect the temperature dependence of a superconducting volume.

### 3.2.2. Results of Measurement and Calculation

Typical $\chi(T)$ curves of studied samples are shown in Figure 4a. Blue curves demonstrate the susceptibility of a bulk Pb sample with a size of 1 mm. The real part $\chi'(T)$ indicates an abrupt superconducting transition at $T_C$=7 K with a width of 0.1 K. The red curves show the susceptibility of the Pb–In 65/35 sample shown in Figure 2a. The real part $\chi'(T)$ (dashed curve in a picture) indicates a superconducting transition: $\chi'(T)$ starts to decrease at T = 6.8 K and an abrupt change occurs at T = 6.1–6.2 K. Starting at this temperature, the imaginary part $\chi''(T)$ demonstrates a considerable absorption of the electromagnetic field in the sample. Similar temperature dependencies of susceptibility were also obtained for samples of Pb–In 76/24 and Pb–In 99/1, indicating that the temperature of the superconducting transition of the samples does not significantly depend on the indium content. In contrast, in bulk Pb–In alloys, the transition temperature $T_C$ depended on the In concentration in an alloy [43]. Our materials are polycrystalls consisting of crystals with different concentration of indium, and such concentration is significantly more than concentration in earlier described materials. Black and green curves present the susceptibility in Pb–In 85/15 (Figure 2b) and Pb–In–Pd 93/4/3 (Figure 2c) samples, respectively. The real part $\chi'(T)$ does not show an abrupt change under superconducting transition. Moreover, with the accuracy of the experiment, the absorption is not present. This behavior is explained within a model of uncoupled superconducting domains with sizes on the order of the depth of the magnetic field penetration in the superconductor at this temperature, as in [35].

Using the temperature dependence of the magnetic field penetration depth for pure Pb with $\lambda(0)$ = 40 nm and Equation (1), we estimated the superconducting domain size to be 300 nm in Pb–In 85/15 and Pb–In–Pd 93/4/3 samples. The estimated domain size from the Pb–In 65/35 sample curve was found to be 5 times larger, that is, 1.5 μm. From Figure 2, one can see that the obtained superconducting domain sizes in Pb–In 85/15 samples are comparable with the size of the second level of the structural hierarchy, i.e., to the crystallite sizes. Thus, the data suggest that the screening supercurrent flows freely between nanocrystals, but the supercurrent is strongly suppressed or even absent between the crystallites. On the other hand, the obtained result indicates that an electrodeposition on templates by the programmed pulsing current may serve as an effective tool to control the size of the superconducting domains in 3D metallic nanostructures.

Finally, Figure 4b presents the temperature dependence of the real part of the dynamic susceptibility in a small constant magnetic field of 150 Oe. The blue curve shows the susceptibility of the bulk Pb sample. The red curve shows the susceptibility of the Pb–In 65/35 sample. These two curves demonstrate common features of the effect of the magnetic field on susceptibility: a reduction in the temperature of the superconducting transition and a considerable change in the temperature dependence of susceptibility $\chi'(T)$. The black and green curves represent the temperature dependence of susceptibility of the Pb–In 85/15 and Pb–In–Pd 93/4/3 samples, respectively, in the magnetic field of 150 Oe. One can see that in these samples, the magnetic field does not significantly affect both the superconducting transition and the form of the temperature dependence of the susceptibility. Such behavior is expected within the model of uncoupled superconducting domains with a size smaller than $\lambda(T)$ [35].

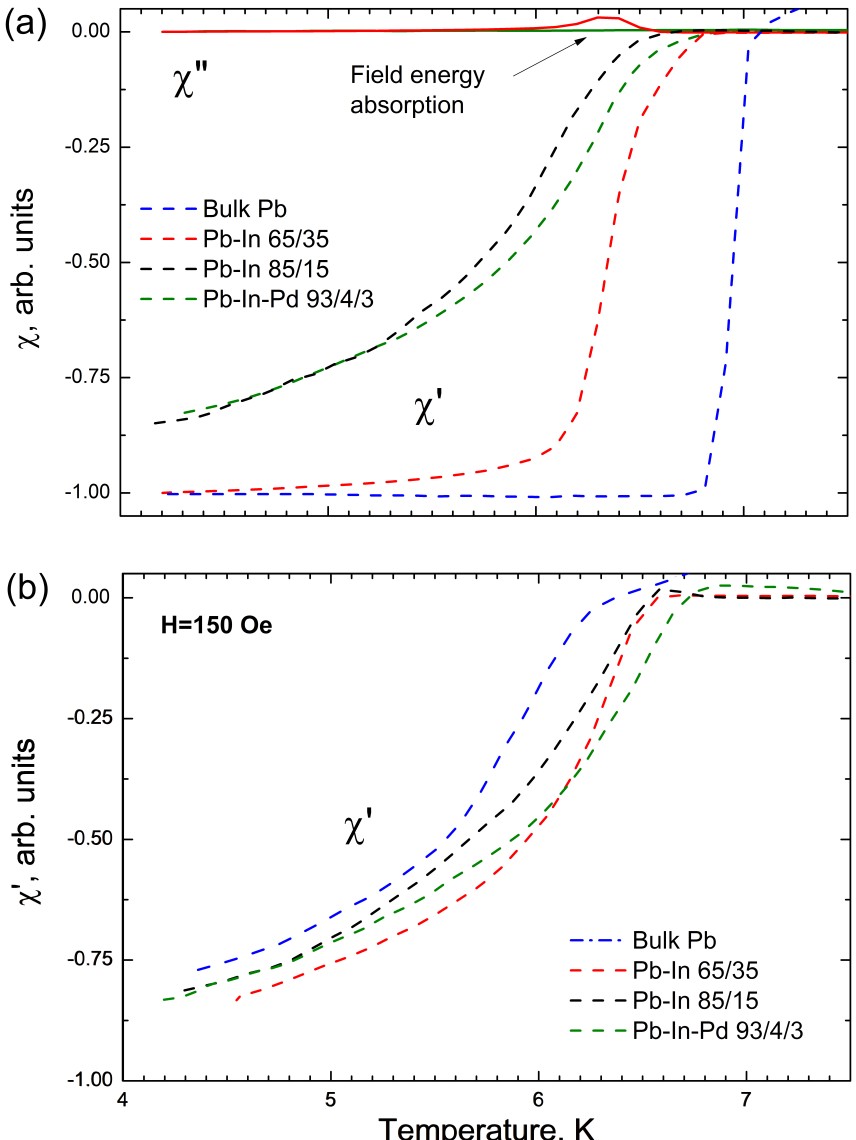

**Figure 4. Temperature dependence of magnetic susceptibility $\chi(T) = \chi'(T) + i\chi''(T)$ of different samples**. Dashed lines: $\chi'(T)$, solid lines: $\chi''(T)$. (**a**) $\chi(T)$ for samples without magnetic field: bulk Pb, Pb−In 65/35 sample shown in Figure 2a, Pb−In 85/15 and Pb−In−Pd 93/4/3 samples shown in Figure 2b,c. (**b**) $\chi'(T)$ for samples in the magnetic field of $H = 150$ Oe: bulk Pb, Pb−In 65/35 sample shown in Figure 2a, Pb−In 85/15 and Pb−In−Pd 93/4/3 samples shown in Figure 2b,c. Only real part $\chi'(t)$ is shown.

## 4. Conclusions

To summarize, we synthesized hierarchical structures of the Pb–In alloy obtained via the pulsing current electrodeposition on porous membranes. Careful control of the synthesis parameters allowed us to find the optimal conditions for growing two types of structures are synthesized: chains with a length of several tens of micrometers consisting of crystallites with a size of 1 µm, containing 50–70 nm nanocrystals, and rods with a length of several micrometers, consisting of 300 nm crystallites containing 10–20 nm nanocrystals.

Temperature dependences of dynamic magnetic susceptibility of the nanostructured chains (Pb–In 65/35) show an abrupt superconducting transition at 6.4 K with a width of 0.3 K in their real part $\chi'(T)$ and a significant absorption in their imaginary part $\chi''(T)$ under the transition from the normal to the superconducting state. On the contrary, the temperature dependence $\chi'(T)$ of Pb–In 85/15 nanorods and Pb–In–Pd 93/4/3 samples does not demonstrate the abrupt superconducting transition and, with instrumental preci-

sion, the absorption of $\chi''(T)$ is absent under the transition. A comparison of the results with a theoretical model for the Pb–In 85/15 sample yields the characteristic sizes of uncoupled superconducting domains, which are found to be compatible with the sizes of the crystallites (1.5 μm).

We also used the X-ray technique and microscopy techniques in our work, which is a common method for studying 3D hierarchical nanoscaled materials. However, such methods do not allow to investigate electrical contacts between objects. We used a noncontact method, a measurement of the magnetic susceptibility, which allowed us to reach the clear conclusion that the nanocrystallites in our samples that were in the superconducting state were electrically connected to each other.

We hope that the results of this article may be useful for the synthesis of complicated dielectric, metal, or superconducting nanocrystals with multilevel organization, in the development of new superconducting nanomaterials, for the investigation of electrical conductivity in crystals, and for the investigation of nanoscale size effects in 0D, 1D, 2D, and 3D structures.

**Author Contributions:** Synthesis of nanostructures and microscopy techniques, G.K.S and G.V.S.; X-ray technique, I.M.S; visualization, D.S.Y. and I.A.N.; Dynamic Magnetic Susceptibility, D.V.S. and A.F.S.; data analysis and writing original draft preparation A.F.S., G.K.S., I.M.S., G.V.S., S.A.V., D.S.Y, I.A.N. and D.V.S. writing review and editing D.S.Y, I.A.N. and A.F.S. All authors have read and agreed to the published version of the manuscript.

**Funding:** Rosatom (Russia): Contract No. 868-1.3-15/15-2021

**Data Availability Statement:** The data presented in this study are available on request from the corresponding author.

**Acknowledgments:** D.S.Y. and I.A.N. acknowledges support by the Rosatom in the framework of the Roadmap for Quantum computing.

**Conflicts of Interest:** The authors declare no conflict of interest.

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
