# Peer review of "Superconductivity in Hierarchical 3D Nanostructured Pb–In Alloys"

_symmetry, doi:10.3390/sym14102142_

Round 1
Reviewer 1 Report
In this manuscript the authors report on the synthesis and measurement of nanostructured Pb-In and Pb-In-Pd alloys. They consider different alloy fractions and also vary the growth parameters. This results in nanostructured samples with different structures and sizes.
They characterize the samples with SEM, XRD, and use AC susceptibility to measure the superconducting properties. From the XRD they conclude that the particles have a core-shell structure, where the shell of the nanoparticles has a different lattice constant but isomorphic structure.
Overall the paper is well written and the results are interesting. The presentation of the data on superconductivity however could be improved, and the authors may want to rewrite these sections. Some questions are:
1. It would be useful to state at the beginning what effect In (and Pd)
has on the Tc of Pb in the bulk.
2. There is some confusion in Fig 4: There is one green curve that is labeled "Pb-In 85/15, Pb-In-Pd 93/4/3". It seems unlikely that the data for these were identical, or is a line missing? Line 201-202 says green "curves" (plural).
Finally, I noticed some typos:
1. Fig 2(c) heading should be "Pb-In-Pd"
2. Fig 4(a) "absopbtion" should be "absorption"
3. Line 240-241: Shouldn't it be 65/35 that shows the abrupt SC transition and absorption, not 85/15?
Author Response
(Reviewer 1)
In this manuscript the authors report on the synthesis and measurement of nanostructured Pb-In and Pb-In-Pd alloys. They consider different alloy fractions and also vary the growth parameters. This results in nanostructured samples with different structures and sizes.
They characterize the samples with SEM, XRD, and use AC susceptibility to measure the superconducting properties. From the XRD they conclude that the particles have a core-shell structure, where the shell of the nanoparticles has a different lattice constant but isomorphic structure.
Overall the paper is well written and the results are interesting. The presentation of the data on superconductivity however could be improved, and the authors may want to rewrite these sections. Some questions are:
We appreciate the comment very much. We rearranged the manuscript according to the Reviewer requirements.
1. It would be useful to state at the beginning what effect In (and Pd)
has on the Tc of Pb in the bulk.
We absolutely agree with the reviewer that If we talk about a bulk material, then big inclusions of nonmagnetic impurities in a superconductor will lead to a change in the temperature of the superconducting transition. The reference [Rao, C.; Dubeck, L.; Rothwarf, F. Superconducting energy gaps from magnetization measurements: Pb-In system. Physical Review B 1973, 7, 1866.87] has been added to the manuscript.
2. There is some confusion in Fig 4: There is one green curve that is labeled "Pb-In 85/15, Pb-In-Pd 93/4/3". It seems unlikely that the data for these were identical, or is a line missing? Line 201-202 says green "curves" (plural).
We are sorry for the line missing in Fig 4. We double checked the scale bars and addea new black line which is responsible for Pb-In 85/15. Thank you very much for the remark.
Finally, I noticed some typos:
1. Fig 2(c) heading should be "Pb-In-Pd"
Thanks again. Done.
2. Fig 4(a) "absopbtion" should be "absorption"
Done
3. Line 240-241: Shouldn't it be 65/35 that shows the abrupt SC transition and absorption, not 85/15?
Done
Reviewer 2 Report
In the presented work entitled „Superconductivity in hierarchical 3D nanostructured Pb-In alloys” by A. Schevchun et al., the Authors studied superconducting properties of the Pb-In allows by the measurement of dynamic susceptibility χ(T) temperature dependence. They observed various behavior of the χ(T) function and found that it depends on the ratio between a superconducting domain size and a crystallite size. As a result, they suggest that the obtained results paves the way to control a superconducting domain size in materials with the parameters of a pulsed current.
In general, the presented manuscript is written in a clear and well organized manner, while the analysis seems to be free of major errors. Therefore, I would like to recommend present paper for publication after the following corrections (which I believe should improve the paper even further). In particular:
1. The introduction is written in a well-structured and interesting manner, but still has a room for improvement. Of special importance is the discussion of the scale-effects in terms of the superconducting materials. I believe that it may be instructive for the readers to refer to few other cases where scale-effects play an important role, instead on directly moving to the 0D systems (see the first paragraph). This aspect is especially visible in comparison to the later part of the introduction where the Authors already expand their discussion on other low-dimensional superconductor in a form of 2D thin films, nanowires and nanoparticles. In what follows, I suggest to refer to the 2D and 1D systems in the first paragraph to better build the context and uniform discussion in the introduction. The Authors may mention there already cited studies on the 2D and 1D materials. Nevertheless, I also turn the Authors attention to other noteworthy examples of size effects in low-dimensional superconductors which may be included in their introduction. In particular, please see the decrease of Tc (practically a suppression of the superconducting state) when comparing bulk CaC6 with its 2D monolayer counterpart and in contrary the increase of Tc (practically an induction of superconductivity) when comparing 2D monolayer LiC6 and its bulk form (for discussion of both materials please refer to Nature Physics 8, pages 131-134 (2012)). In reference to the latter LiC6 and the already mentioned influence of size effects on the crystal lattice and electrons the Authors may also refer to the recent studies on the non-adiabatic superconductivity in LiC6 as caused by the comparable electron and phonon energy scales at low-dimension (see Physical Review B 99 (22), 224512 (2019)). As for the 1D materials I strongly recommend review by Arutyunov et al. (see Physics Reports 464, 1-70 (2008)). To this end, I also suggest to refer to the relatively recent review of size-effects in 0D materials (please see: Reports on Progress in Physics 77 116503 (2014)).
2. Additionally, the 4th paragraph of the Introduction may be improved in my opinion for better readability. Kindly merge it with the last 5th paragraph and improve the first sentence to better refer to the previous text. Currently the paragraph feels detached from the rest of the text on the first reading.
3. In relation to the two above points I cannot understand what is the motivation of the Authors to consider the superconducting properties of the 3D hierarchical nanoscaled materials in the first place. Please provide exact rationale.
4. In my opinion, the sentence in abstract starting “Two different kinds of χ(T) dependence were observed in synthesized structures …” is too lengthy. I believe that two shorter sentences will be much better in terms of readability. Also, please try not too overuse “and” word in this new sentences and try to replace it with some synonyms.
5. The “theory of uncoupled domains” is mentioned multiple times in the text, however, without any reference or even brief explanation. Please provide some description of this theory and potential references for less experienced readers.
6. The equation (1) may use a full stop (dot) at the end. Also, why the Authors suddenly omit temperature dependence when writing χ’ in Eq. (1)?
7. The description of Fig. (4) is confusing. The y-axis is described as χ’, while in the main text and the figure’s caption the Authors describe it as the dynamic susceptibility (which reads χ according to line 170). Is this the dynamic susceptibility or its real part? Please clarify. Also I don’t quite understand the meaning of the real part as given in lines 173-174, is there some typo?
8. Unfortunately, to some extent the conclusions just repeat description of the obtained results. I would like to encourage the Authors to improve this section so the Readers can get some more new insight from their study. For example please extend the last paragraph of this section in this context.
*end of report*
Author Response
(Reviewer 2)
In the presented work entitled „Superconductivity in hierarchical 3D nanostructured Pb-In alloys” by A. Schevchun et al., the Authors studied superconducting properties of the Pb-In allows by the measurement of dynamic susceptibility χ(T) temperature dependence. They observed various behavior of the χ(T) function and found that it depends on the ratio between a superconducting domain size and a crystallite size. As a result, they suggest that the obtained results paves the way to control a superconducting domain size in materials with the parameters of a pulsed current.
In general, the presented manuscript is written in a clear and well organized manner, while the analysis seems to be free of major errors. Therefore, I would like to recommend present paper for publication after the following corrections (which I believe should improve the paper even further). In particular:
We appreciate the comment very much. We rearranged the manuscript according to the Reviewer requirements.
- The introduction is written in a well-structured and interesting manner, but still has a room for improvement. Of special importance is the discussion of the scale-effects in terms of the superconducting materials. I believe that it may be instructive for the readers to refer to few other cases where scale-effects play an important role, instead on directly moving to the 0D systems (see the first paragraph). This aspect is especially visible in comparison to the later part of the introduction where the Authors already expand their discussion on other low-dimensional superconductor in a form of 2D thin films, nanowires and nanoparticles. In what follows, I suggest to refer to the 2D and 1D systems in the first paragraph to better build the context and uniform discussion in the introduction. The Authors may mention there already cited studies on the 2D and 1D materials. Nevertheless, I also turn the Authors attention to other noteworthy examples of size effects in low-dimensional superconductors which may be included in their introduction. In particular, please see the decrease of Tc (practically a suppression of the superconducting state) when comparing bulk CaC6 with its 2D monolayer counterpart and in contrary the increase of Tc (practically an induction of superconductivity) when comparing 2D monolayer LiC6 and its bulk form (for discussion of both materials please refer to Nature Physics 8, pages 131-134 (2012)). In reference to the latter LiC6 and the already mentioned influence of size effects on the crystal lattice and electrons the Authors may also refer to the recent studies on the non-adiabatic superconductivity in LiC6 as caused by the comparable electron and phonon energy scales at low-dimension (see Physical Review B 99(22), 224512 (2019)). As for the 1D materials I strongly recommend review by Arutyunov et al. (see Physics Reports 464, 1-70 (2008)). To this end, I also suggest to refer to the relatively recent review of size-effects in 0D materials (please see: Reports on Progress in Physics 77 116503 (2014)).
Thanks for the comment. We have rewritten the introduction. We have added the following text and corresponding references to the introduction (lines 15-23 and 62-76):
Zero-dimensional (0D systems) [1], one-dimensional (1D systems) [2], or two-dimensional (2D systems) [3] dimensions are the low-dimensional systems. Quantum dots, quantum wells, and quantum nanowires are the popular names for the 0D, 1D, and 2D structures in the Society of Condensed matter and Solid-State Physics, respectively. The usage of closed structures is also common in many other scientific domains, including thermoelectric materials [4], metamaterials [5], topological materials [6], energy storage applications [7], and catalysis [8]. Since confinement typically takes place at the nanoscale scale in these applications, low-dimensional systems are typically referred to as nanostructures.
For example, the novel class of two-dimensional hexagonal superconductors show a drop in TC of 2D monolayer CaC6 is compared to its bulk equivalent, but an increase in TC when monolayer LiC6 is compared to its bulk form [27] . Materials’ critical temperature values are influenced by non-adiabatic effects due to similar electron and phonon energy scales at low dimensions. In particular, the other materials might react similarly to LiC6 and subsequently have their TC reduced as a result [28]. As a result, the articles mentioned above help us comprehend the underlying physics of low-dimensional superconductivity and present new avenues for further study. This brief review indicates that low-dimensional superconductors like thin films, nanowires, and nanopowders serve as convenient objects for studying the effect of size on superconductivity. Recently a novel class of materials has been developed containing hierarchical 3D nanostructures [29 ] with complex symmetry nano-architecture [30–32 ].
- Additionally, the 4th paragraph of the Introduction may be improved in my opinion for better readability. Kindly merge it with the last 5th paragraph and improve the first sentence to better refer to the previous text. Currently the paragraph feels detached from the rest of the text on the first reading.
Thank you very much for the remark. In the new version of the text we combined 4th and 5th paragraph. We tried to improve these sections for better readability.
- In relation to the two above points I cannot understand what is the motivation of the Authors to consider the superconducting properties of the 3D hierarchical nanoscaled materials in the first place. Please provide exact rationale.
We hope that the results of this article may be useful for the synthesis of complicated dielectric, metal, or superconducting nanocrystals with multilevel organization, in the development of new superconducting nanomaterials, for the investigation of electrical conductivity in crystals, and for the investigation of nanoscale size effects in 0D, 1D, 2D, and 3D structures.
In addition to 1D metallic and carbon nanostructures such as nano and quantum wires, nanotubes, fibers, cones, and tips, which are highly promising classes of building blocks for a variety of established and developing nanoscale systems and functional elements in optoelectronic, energy, and biomedical devices, 3D hierarchical nanoscale materials are also receiving a lot of attention from both academia and business.
- In my opinion, the sentence in abstract starting “Two different kinds of χ(T) dependence were observed in synthesized structures …” is too lengthy. I believe that two shorter sentences will be much better in terms of readability. Also, please try not too overuse “and” word in this new sentences and try to replace it with some synonyms.
Thanks for the proposal. We are sorry for the too overuse “and” words. We double checked the “and” words in our sentences and try to replace it.
- The “theory of uncoupled domains” is mentioned multiple times in the text, however, without any reference or even brief explanation. Please provide some description of this theory and potential references for less experienced readers.
Thank you for your valuable comment. The parameter depends on lambda(T) (penetration depth at a given temperature), and not lambda(0). The weak dependence of the susceptibility on the magnetic field in this case is just due to the fact that the particle (domain) already has a field, its size is smaller than \lambda(T). The reference [Neminsky, A. M., et al. "Temperature dependence of the magnetic field penetration depth in Rb 3 C 60 measured by ac susceptibility." Physical review letters 72.19 (1994): 3092.] has been added to the manuscript. Line 228
- The equation (1) may use a full stop (dot) at the end. Also, why the Authors suddenly omit temperature dependence when writing χ’in Eq. (1)?
Done
- The description of Fig. (4) is confusing. The y-axis is described as χ’,while in the main text and the figure’s caption the Authors describe it as the dynamic susceptibility (which reads χ according to line 170). Is this the dynamic susceptibility or its real part? Please clarify. Also I don’t quite understand the meaning of the real part as given in lines 173-174, is there some typo?
We are sorry for the line missing in Fig 4. We double checked the scale bars and addea new black line which is responsible for Pb-In 85/15. We changed figure captions accordingly. Thank you very much for the remark.
- Unfortunately, to some extent the conclusions just repeat description of the obtained results. I would like to encourage the Authors to improve this section so the Readers can get some more new insight from their study. For example please extend the last paragraph of this section in this context.
Thank you for your valuable comment.
We have completely revised the conclusions of our article.